# Design and Implementation of a Collaborative Clinical Practice and Research Documentation System Using SNOMED-CT and HL7-CDA in the Context of a Pediatric Neurodevelopmental Unit

**DOI:** 10.3390/healthcare11070973

**Published:** 2023-03-29

**Authors:** Bruno Direito, André Santos, Susana Mouga, João Lima, Paulo Brás, Guiomar Oliveira, Miguel Castelo-Branco

**Affiliations:** 1Coimbra Institute for Biomedical Imaging and Translational Research (CIBIT), Institute of Nuclear Sciences Applied to Health (ICNAS), University of Coimbra, 3000-548 Coimbra, Portugal; 2Instituto do Ambiente, Tecnologia e Vida, 3000-214 Coimbra, Portugal; 3Coimbra Clinical Academic Center, Faculty of Medicine, Coimbra University Hospital, Pediatric Hospital, 3000-602 Coimbra, Portugal; 4Child Developmental Center, Research and Clinical Training Center, Hospital Pediátrico, Centro Hospitalar e Universitário de Coimbra, 3000-602 Coimbra, Portugal; 5University Clinic of Pediatrics, Faculty of Medicine, University of Coimbra, 3000-548 Coimbra, Portugal

**Keywords:** healthcare documentation, electronic medical records, medical research, autism spectrum disorder, longitudinal medical records, data harmonization, disease classification

## Abstract

This paper introduces a prototype for clinical research documentation using the structured information model HL7 CDA and clinical terminology (SNOMED CT). The proposed solution was integrated with the current electronic health record system (EHR-S) and aimed to implement interoperability and structure information, and to create a collaborative platform between clinical and research teams. The framework also aims to overcome the limitations imposed by classical documentation strategies in real-time healthcare encounters that may require fast access to complex information. The solution was developed in the pediatric hospital (HP) of the University Hospital Center of Coimbra (CHUC), a national reference for neurodevelopmental disorders, particularly for autism spectrum disorder (ASD), which is very demanding in terms of longitudinal and cross-sectional data throughput. The platform uses a three-layer approach to reduce components’ dependencies and facilitate maintenance, scalability, and security. The system was validated in a real-life context of the neurodevelopmental and autism unit (UNDA) in the HP and assessed based on the functionalities model of EHR-S (EHR-S FM) regarding their successful implementation and comparison with state-of-the-art alternative platforms. A global approach to the clinical history of neurodevelopmental disorders was worked out, providing transparent healthcare data coding and structuring while preserving information quality. Thus, the platform enabled the development of user-defined structured templates and the creation of structured documents with standardized clinical terminology that can be used in many healthcare contexts. Moreover, storing structured data associated with healthcare encounters supports a longitudinal view of the patient’s healthcare data and health status over time, which is critical in routine and pediatric research contexts. Additionally, it enables queries on population statistics that are key to supporting the definition of local and global policies, whose importance was recently emphasized by the COVID pandemic.

## 1. Introduction

Improving healthcare services depends heavily on integrity, reliability, accuracy, interoperability, and high-quality general information about the patient [1]. The development of electronic health (e-health) and electronic health record (EHR) systems that address these features is essential to delivering high-quality care.

According to the definition of the International Organization for Standardization, “the EHR is a repository of information regarding the health of a subject of care in computer processable form, stored and transmitted securely, and accessible by multiple authorized users. Its primary purpose is the support of continuous, efficient, and quality integrated healthcare, and it contains information, which can be historical, current, and expected” [2]. The longitudinal, comprehensive characteristic of the EHR, combined with the amount of stored data among patients, favors transforming clinical data into knowledge to improve care.

Semantic interoperability (SI) is, at present, a crucial concept in healthcare, particularly in the development of EHR. Each system must be considered a node in a complex network of local, national, and international healthcare services. Lack of information coordination and communication between health systems hinders the prompt application of appropriate measures [3].

The definition of SI is not trivial. The European Commission defines SI as “the ability of information and communication technology (ICT) systems and the business processes they support to exchange data and to enable sharing of information and knowledge”. The International Organization for Standardization (ISO) EN13606 on semantic interoperability refers to “the ability to automatically interpret the information exchanged meaningfully and accurately in order to produce useful results as defined by the end users of both systems”.

In healthcare documentation, two main complementary frameworks support EHR development towards the goal of SI, namely information model and terminology [2]. The information model represents the standard structure for the information and the set of rules concerning the organization of clinical concepts. Terminology in healthcare refers to a group of structured symbolic representations of the meaning and context of clinical ideas. Clinical terminologies are essential in developing EHR systems, anchoring automation in acquiring, coding, processing, analyzing, and communicating clinical information. Several authors discuss that using clinical terminologies in electronic records leads to better quality information [2,4,5,6]. There are several messaging standards, content standards, and clinical terminologies available and implemented in e-health systems, including the Continuity of Care Document (CCD) [7] specification, the Health Level Seven (HL7) messaging standard, the HL7 clinical data architecture (CDA) [8], the HL7 EHR functional model, Logical Observation Identifiers Names and Codes (LOINC) [9], and SNOMED Clinical Terms (SNOMED CT) [10].

The continuous improvement and increased adoption of EHRs anticipate the availability of an unprecedented amount of data [11]. One of the most critical healthcare challenges is the ability to process and extract data from EHRs to provide knowledge to the medical and scientific communities. In this sense, the development and application of data analysis methods in EHRs are emerging and will ultimately transform EHRs into learning e-health systems [12].

### 1.1. Background and Related Work

Previous EHR systems were developed with explicit logic, i.e., EHR systems were based on a predefined encoding dictionary, limiting the medical domain knowledge. This rigidity imposes several constraints. Changes in this explicit knowledge domain depend on additional costs and iterations with vendor and software development teams. Multidisciplinary, multi-site collaboration, and data sharing are also hindered because of the different requirements among groups or medical specialties. Additionally, the definition of team-specific features challenges the achievement of semantic and logical interoperability.

One of the most relevant solutions at the level of semantic interoperability is the openEHR standard. The main feature of the openEHR design is how clinical concerns and technical development are separated. On the one hand, a reference model addresses the technical considerations, particularly information structure and data types. On the other hand, clinical concerns deal with specific domain semantics. Together, they allow the composition of larger structures called templates.

Novel solutions should be flexible to allow users to conduct the following: i. maintain all definitions with minimal intervention from software development teams; ii. link elements from external terminologies; iii. implement a common standard to facilitate data transfer and re-usability; and iv. to avoid data duplication, i.e., integration with other local EHR systems [5,13,14,15]. Additionally, EHR solutions should increase efficiency in data acquisition, improve data structuring, and ultimately support the development of clinical decision support systems (CDSS) [16,17].

Several authors focused on semantic and document structure interoperability based on SNOMED CT (used as a clinical terminology) and the HL7-CDA documentation standard. We performed a literature search with the following query: *(“Electronic Medical Record” OR EMR OR “Electronic Health Record” OR EHR OR “Computerized Medical Record” OR CMR OR “Automated Medical Record” OR AMR OR “Hospital Information System” OR “Health Information System” OR HIS OR “Clinical Information System” OR CIS OR “Medical Record System”) AND Interoperability AND (HL7 CDA) AND (SNOMED or SNOMED CT)*. This query was performed on the following databases: Google Scholar; IEEE Xplore; Nature; PubMed; Science Direct; Scopus; and SpringerLink. Articles included in the analysis met the following criteria: (1) peer-reviewed and full-length articles written in English and (2) published between 2012 and 2022 in the selected databases. We excluded studies related to PHR integration with EHR, search engines for ontology-based semantics, and theoretical works. Thus, our search resulted in a sample of peer-reviewed and full-length studies on EHR architecture operating in hospital/clinical information systems using healthcare standards focused on interoperability.

Shanbehzadeh et al. (2021) [18] developed an interoperable framework for public health monitoring. They focused on semantic sharing and collaborative modeling that met the information exchange requirements of a surveillance system of notifiable diseases. The implementation required a selection of data fields determined according to a literature review and field experts, which were matched to SNOMED-CT and a report form based on HL7-CDA. The result was a minimum dataset report structure that offers an inclusive and interoperable dataset. These features help make notifiable disease data more comparable and reportable across studies and organizations.

Oliveira et al. (2020) [19] implemented an OpenEHR-based system that allowed the organization of clinical information using valid and reusable clinical structures. The system supports the creation of operational templates and dynamically generates data based on the templates defined.

Another critical aspect addressed using interoperable reporting platforms was post-market drug surveillance. Declerck et al. (2015) [20] aimed to record drug-related adverse events (AEs) to facilitate and accelerate the reporting process. Standard HL7 CDA and proprietary EHR models were mapped to a standard information model. Furthermore, they developed an automatic conversion system to transform terminologies used in the EHR (e.g., International Classification of Diseases, 9th Revision, Clinical Modification (ICD-9-CM), International Classification of Diseases 10th revision (ICD-10), LOINC, or SNOMED CT) to the target terminology (MedDRA) expected in AE reporting forms.

López-Nores et al. (2012) [21] developed a system to promote daily life medication adherence by supporting and combining information from domestic and mobile devices’ EHR repositories. The system was designed to enforce medical prescriptions and issue user warnings. By gathering and processing user prescription information from EHRs, the system reacts to potential medication non-adherence. The implementation adopted the HL7 standard to define protocols to exchange healthcare messages, the openEHR to represent content and structure data entities in the EHR, and the EN13606 standard to ensure privacy and security. Highlights included automatic download of medical prescriptions, warnings to avoid forgetting the medicine when leaving home, buying suitable foods (e.g., lower cholesterol levels) when entering supermarkets, and buying new doses of nearly depleted drugs when passing a drug store.

These works also highlight several limitations of such platforms. Identifying the correct information at the insertion moment is critical to data acquisition automation. Error detection and raising warnings if the data value is out of the predefined range may minimize the number of errors. Large free-text components still limit automatic knowledge extraction from reports. The need for a common standard and mapping between terminologies also represent significant obstacles. One major issue is the engagement of healthcare agents, as tasks related to creating templates and mapping components from terminologies are often time consuming and considered with no significant advantages to daily activities.

### 1.2. Objectives

This document aims to present an extension to the EHR solution used in the National Health Service of Portugal (SNS) by adding structured clinical documents. To this end, the proposed platform combines the HL7 CDA standard and the clinical terminology of SNOMED CT. This solution is integrated with the SClínico [22], a healthcare informatics platform provided by SNS, to assist nearly 66.500 public healthcare agents in standardizing clinical record procedures. It aims to overcome the limitations imposed by previous in-place documentation strategies, mainly based on free-form text, that severely constrain electronic storage, transmission, standardization, and the re-use of information for developing CCDS and research purposes.

In the Neurodevelopment and Autism Unit (UNDA) of the University Hospital Center of Coimbra (CHUC), the multidisciplinary, longitudinal view of data are particularly hindered. The previous approach included the following: i. a patient-specific single file to record longitudinal data, edited multiple times over time; ii. unstructured data acquisition during clinical encounters; and iii. concurrent access to files (especially relevant in multidisciplinary teams, i.e., patients meet different elements from our clinical team in a single visit to UNDA).

To implement and validate the new solution, a multidisciplinary task force involving the development team, healthcare, and research end users was created in UNDA. Together, we developed a set of tailored templates that ease data acquisition in the clinical context, both from cross-sectional and longitudinal points of view, while maintaining compliance with the best practices previously established.

Supported by an international medical terminology standard, the new solution focuses on transparent clinical data coding and structuring, as well as preserving data quality and interoperability. Based on the development of coded templates (selected according to preferred clinical terms or expressions), the approach aims to support semantic interoperability facilitating the encoding (using the SNOMED-CT terminology and HL7 CDA document structure) of clinical data information during data acquisition while simultaneously establishing a common platform between clinical practice and research. The system proposed also addresses essential aspects of national and European legislation on data privacy. The main objective is to create an information system that is sufficiently generic to address multiple targets and respond to different limitations. As a complex and highly interdisciplinary use case, we applied and developed the framework in close collaboration with UNDA. We assessed our approach following the main functionalities proposed in [23,24].

## 2. Materials and Methods

### 2.1. Design Considerations and Key Requirements

One of the critical requirements of the proposed system is the ability to appropriately record and document data from routine clinical encounters, here considered as a single event in which care is given, most importantly to support knowledge extraction from these data. To avoid redundancy with local EHR systems, the solution integrates with the systems currently operating in the clinical environment (compliant with national healthcare standards), i.e., to avoid duplication of previously inserted data. To optimize data insertion, users, based on their expertise, should be allowed to define a set of data items (selected and linked to a terminology service) and record them during the clinical encounter. The users should be able to set restrictions, including data type and valid ranges, easing the data insertion process. This set of features is the backbone of larger structures called templates and should allow different data organization, optimizing the layout’s filling during data insertion.

The design of templates should be possible without the intervention of third-party experts or software developers, i.e., the interface should be flexible enough to allow the clinical teams to create the appropriate templates.

These conceptual templates should be shared and re-usable beyond the original author so that semantics are maintained to the utmost.

Ultimately, the proposed framework minimizes large free-text blocks and promotes the encoding of data fields in the recording moment. The resulting datasets increase technical and semantic interoperability and allow automated interpretation and clustering of data.

Additionally, a subset of standard functional components in HL7 EHR-S FM [23,24] was adopted in this study to ensure the quality of interoperability and EHR implementation standards (Table 1). The HL7 EHR-S FM standard provides a well-defined functional profile for healthcare systems.

### 2.2. Framework Architecture

The basic architecture of the extension to the EHR is introduced here. Three-layered architecture separates three distinct functionalities: data storage, logic, and user interface. These are based on three layers: data access, logic, and presentation. This modular implementation reduces dependencies between components and yields efficient development, maintenance, scalability, and security (Figure 1). Furthermore, the structure is connected to a SNOMED CT terminology service, an essential part of the semantic interoperability strategy. The main purpose of this is to yield coded templates that ultimately enable the encoding of data during data insertion in the clinical encounter.

#### 2.2.1. Database Layer

The data access layer implements two types of routines: i. to connect to a database managed by MS SQL server, where the core data of the solution are stored; ii. to access SONHO V2 (the Portuguese national database responsible for supporting the administration of hospital services integrated with the national health service, SNS), where administrative data are available.

To access the data in the MS SQL server-managed database, we implemented a solution based on the entity framework version 4.5 [25]. This framework is an object-relational mapper (ORM) that enables the use of domain-specific objects, effortlessly handling the create, retrieve, update, and delete (CRUD) operations. Using the repository pattern, we can create an abstraction layer between the data access layer and the business logic layer that facilitates access to backend components. It also allows persistent maintenance of data derived from other hospital EHR systems.

#### 2.2.2. Logic Layer

The logic layer contains the services necessary to perform CRUD operations, i.e., it represents the interface between the user view and the data storage. It is implemented as a representational state transfer (REST) service due to its scalability and the ease in which it can exchange of information between the databases and the presentation layer. This architectural style involves building resource-oriented architecture (ROA) by defining resources that implement uniform interfaces using standard HTTP, identified by a uniform resource identifier (URI).

#### 2.2.3. User Interface

The interface layer implements a web-based user layout based on the ASP.NET MVC framework. It enables standard user routines, such as login, search, template creation, access to SNOMED CT terminology, template filling, administrative data consultation, etc.

The layer comprises the three components inherent to the models, views, and controllers (MVC) frameworks. This approach enables clean separation between the data models, controllers, and views.

The model component represents the data structure and intends to replicate the data and data format available in the data access layer via the web services available.

The view component consists of all the user interface design. The user interface (UI) is implemented in HTML and JavaScript. It uses the native ASP.NET MVC 4 capabilities (i.e., Razor) to access data from the models and several JavaScript and CSS frameworks that improve customization, responsiveness, and document creation abilities (e.g., bootstrap, jQuery, jstree, pdfMake, etc.).

The controller component manages the interaction between the UI and the data. This component has functions that access, edit, and transform data. It embeds the interaction between the UI and the web services available to perform CRUD operations.

### 2.3. SNOMED Terminology Server

The terminology is the structural point for the medical domain expertise. The terminology server was developed in a single-layer ASP.NET MVC project. The module has a web-based interface, accessible through a modern browser supporting HTML5, and allows the visualization of the concepts and their relationships. The interface also implements the creation of new concepts and terms and editing features. Based on these routines, we can locally define new concepts and later propose their inclusion in national releases of the terminology.

The user can search for SNOMED-CT concepts during template creation as the main framework connects with the terminology server via API. These concepts, which integrate the subsequent template, must depict the clinical information needed. If non-existent, the user can create custom concepts in the terminology server. Upon creation, new concepts are immediately available in the template editing module.

A local SNOMED-CT reference set is created based on these custom elements. Combining different concepts (medical terms, procedures, questionnaires, etc.) and the ability to create content are critical advantages of SNOMED-CT and represent a fundamental aspect in the context of multidisciplinary clinical teams.

### 2.4. Integration in EHR Context

The solution presented here is available within the SClínico [22]. SClínico is an EHR system provided by SNS and available to clinical teams, which aims to help standardize clinical record procedures. This tool is present in more than fifty entities in the public health sector. Nearly 75% of hospital staff utilize this platform, corresponding to nearly 66.500 public healthcare agents across different sectors. It represents the access point for diverse patient information and supports sharing with multiple healthcare agents while implementing regulatory privacy rules.

Users can create and attach clinical documents regarding clinical encounters using a web interface that presents templates structured according to HL7 CDA encoded with SNOMED-CT concepts, which aligns with the healthcare standardization goals of SClínico.

This option also supports the re-use of data recorded inside the clinical documents in national and international research projects, with significant advantages in longitudinal studies and the selection of cohorts based on specific criteria.

## 3. Results

### 3.1. Template Management

The proposed solution allows users to compose new templates and edit templates previously created. The template is defined according to HL7 CDA (structured information model) and based on SNOMED-CT (clinical terminology). Figure 2 shows the template interface used to build the Griffiths Mental Developmental Scale [26] template, an assessment tool used in clinical practice and research. The structured template may be later used in the patient-centered interface, in which the user can load the structured template and record clinical data related to a specific clinical encounter (filling the template with the appropriate information).

Each template has a structured library of pre-established elements specified in the HL7 CDA information model. The model defines three hierarchical levels: document-level elements, section-level elements, and entry-level elements. Their combination represents specific document types. They contain possible relationships between different HL7 CDA levels (i.e., which sections are valid in each document and which entries are valid in each section).

The interface presents the HL7 CDA elements according to the document type selected and the rules established in the standard specification (e.g., the inclusion of mandatory sections or entries). In this interface, the users can select the document elements according to their requirements without infringing the standard rules.

This structure provides the backbone for logically organized groups of variables. Once the appropriate CDA elements are identified, the user can create conceptual areas and add groups of clinical concepts retrieved from the SNOMED-CT terminology server. The intuitive interface yields the definition of different layouts based on rows and columns (Figure 3).

Once a conceptual area is created, the user may include clinical concepts based on three types of queries (that implement access to the terminology server): i. querying the terminology content specifying a substring; ii. selecting the concept based on the taxonomy representation; and iii. querying by concept identification (id) (requires previous knowledge of the terminology) (Figure 4).

Each area can anchor several editable concepts. Once the concept is selected, the user must set one of five possible data types: numerical text; list; Boolean; and date.

The numerical data type allows the user to define a minimum and maximum data range (these values enable validation during the template filling). The list data type requires the selection of a list of concepts, i.e., possible “answers”. The interface supports, including a string (e.g., instructions), are actionable as a help menu during template filling. Additionally, the user may set any concept as mandatory, preventing document creation unless data are inserted.

The user that created the template is associated with the template in the database. The templates are then saved in the database as an XML schema. The templates can be defined with different accessibility rules: i. personal (only accessible by its creator); ii. associated with the user’s specialty (specialty-specific); and iii. set of specialties, where every registered user can access the template (Figure 5). The XML schema is used to generate the interface for data collection. The template module yields editing of previously created templates.

### 3.2. Patient Context and Clinical Document Management

After user authentication, the system redirects the user to the patient view or dashboard. The patient view displays the patient’s administrative data and general information regarding the current episode (appointment identification, etc.). The authentication process requires information about the user, the patient, and the episode of interest provided by SClinico to assess accessibility rules. This step ensures the user is placed in the proper clinical context. Once in the patient view, the user can start selecting templates and filling documents (Figure 6).

Creating a new document relies on selecting and filling templates from personal, specialty-specific, or open-access templates (Figure 7). A document is associated with an episode, template, and author (corresponding to the user logged in at the time of template filling). Each episode can have as many documents as necessary (based on as many templates as required).

The user interface (UI) is designed according to the XML schemas imported from the selected template. The rules defined in the template are also implemented, i.e., data types, mandatory elements, etc. If the user does not comply with the rules, the document is not saved, and an error message is displayed in the UI (Figure 8).

The document is saved as an XML file if all data are found to comply with the implemented rules. The interface implements the printing of the stored XML files as a portable document format (PDF) file for reporting purposes (these files may function as attachments to other EHR systems) (Figure 9).

An important feature is the ability to edit documents. To this end, the proposed platform can reload the template to create the document and present the previously saved answers. The database implements document version functionality that permits recovery of previous versions of the document and their authors.

Additionally, the user can change the episode date. If there is a match between the episode date and the user permissions to access that episode, access is granted. In this context, previously saved clinical documents can be consulted and edited. Pairing user permissions with episode information allows users to navigate easily through patient episodes and guarantees limited data accessibility.

### 3.3. Clinical Implementation and Use Case

Although the platform’s main objective is to provide a generic information system across multiple specialties, the framework was developed in collaboration with UNDA and focused on autism spectrum disorder (ASD). ASD is an early onset, life-long neurodevelopmental disorder with high worldwide prevalence [27]. It is characterized by deficits in social interaction and communication and a repetitive and limited pattern of behavior and interests [28].

Our clinical setting is Centro Hospitalar e Universitário de Coimbra (CHUC), which integrates a central pediatric hospital (HP) for a population of 2,000,000 inhabitants and pediatric population (0–18 years) about 400,000. The HP includes a child developmental centre (CDC), which integrates a specialized neurodevelopmental and autism unit (UNDA), a national reference for neurodevelopmental disorders, particularly ASD. A comprehensive clinical and biological database of over 6000 patients with neurodevelopmental disorders (more than 2500 with ASD) is available. A large amount of this population is regularly followed multiple times per year. UNDA has an intake of around 300 new cases annually from all over the country.

The gold standard tools for the diagnosis of ASD are used on a daily basis by experienced psychologists and neurodevelopmental pediatricians to establish a clinical diagnosis of ASD. A trained multidisciplinary team works daily to provide clinical, research, and training assistance. The participation of UNDA/CDC/HP/CHUC in national and international research projects is already routine, not only in the genetic area but also in areas such as epidemiology, diagnosis, neuroimaging, and behavioral and cognitive neuroscience [29,30,31,32].

This setting provides a demanding clinical and data science context. Accordingly, manipulating the dataset (creation of new elements, update of stored values, data retrieving) presents several challenges and limitations. The original data recording was performed using FileMaker Pro Software (a cross-platform relational database application from Claris Inc.) and is based on a single view for each patient, i.e., for each patient, an extensive set of elements is recorded in a single file. The completion of the patient’s view is only obtained after several visits. This scenario presents severe limitations and prevents a longitudinal time-dependent perspective of the data, and our project aimed to overcome these limitations in the clinical setting. Because the prior system is not connected to the hospital information system, data duplication is also a limitation. Multidisciplinary teams, in which several individuals add, edit, and delete information, may also be prone to mistakes (use of different unit systems, etc.). The developed framework addresses multiple concerns, such as a longitudinal view of the data, unnecessary data duplication, information coding, and interoperability (key for collaborative, multidisciplinary, multi-institutional studies).

#### Template Development

A task force was created in UNDA to overcome the limitations of prior approaches to data collection. To this end, the multidisciplinary team created a set of templates to ease data acquisition in the clinical context while maintaining compliance with the best practices. The main objectives were identifying and adopting standardized consultation notes guidelines, identifying SNOMED-CT codes and correspondence to the established clinical guidelines, and developing templates for clinical consultation and procedures.

In this sense, the multidisciplinary team (composed of clinicians and other healthcare professionals, such as psychologists, scientists, and developers) created templates for consultation and procedures. The team discussed each component of the templates until obtaining a consensus on concept id, data type, and HL7 CDA element. Team members iteratively refined the templates’ first drafts.

The critical components of the consultation templates are anchored in the assessment section of the HL7 CDA structure. Several conceptual areas were defined within the assessment section as cluster-related concepts to be recorded in the context of the consultation, e.g., neurodevelopmental, adaptive behavior, language, and intellectual assessments, among other domains of interest. The template also organizes information regarding the history of the disease related to ASD within a conceptual area, combining data from pre and perinatal periods (pregnancy and birth history) (Figure 10). This work supported the consolidation and harmonization of terminologies and concepts adopted by the team, which is now ready to migrate to this platform within the clinical academic center.

### 3.4. An Interface with Clinical Research

Clinical research and discovering novel disease diagnoses, treatments, and prognosis approaches depend on information about subjects with a disease of interest, condition, or state. Moreover, the quality of research (and ultimately the quality of the results) also depends on the data quality, structure, and contextualization (metadata).

The system described here addresses the possibility of managing high-quality health data for advanced clinical and research purposes, essential for different research areas, such as the early detection of disease risk, identification of novel biomarkers, patient selection for clinical trials, and patient monitoring or treatment recommendation.

To access the data, the submission of a request form is mandatory. The form indicates the goals of the study, the subset filters (e.g., gender, age range), and the required data (e.g., SNOMED concepts of interest) (Figure 11).

After ethical and data protection guidelines clearance, a dataset is available for download. The datasets have several informative columns (metadata) as follows: patient identifier; the record date; the patient age at the record time; the data required in the form of SNOMED-CT identifiers; the data type; and the value registered (Figure 11). The data are exported in .csv format and offer the possibility for both cross-sectional and longitudinal analysis (see Figure 12).

### 3.5. Achieved her-S FM Functionalities and Challenges

We performed functionality tests to ensure integration with other hospital EHR systems and objectively assessed the implementation of functionalities. Our tests focused on the functional components in EHR-S FM [23,24]. This model has been widely reviewed by healthcare providers, vendors, public health agencies, regulatory and accreditation bodies, professional societies, trade associations, researchers, and other stakeholders, and it expresses consistent EHR system functionality. Table 2 summarizes the EHR-S FM functionalities implemented in the platform.

Considering the main functionalities in Table 2, Table 3 describes the challenges faced while implementing our solution in the EHR hospital-hosted system following the EHR-S FM framework as standard.

## 4. Discussion

### 4.1. Main Findings

To analyze the advantages and limitations of the proposed solution, we performed an analysis of the functionalities models and a comparative analysis with other existing EHR systems. This solution responds to one of the most relevant limitations in current EHR systems in the Coimbra region (Coimbra, Portugal) and still in many clinical sites across Europe, i.e., the inability to store and access structured clinical information and use it in a longitudinal perspective [22,33]. The latter is critical in pediatric settings where health-related data are continuously changing. Furthermore, the definition of templates based on a standard structure (HL7 CDA) coupled with concepts retrieved from known terminology (SNOMED-CT) answered the interoperability challenge because the documents have implicit standardized structure and terminology definitions. Thus, the proposed solution allows healthcare practitioners to create, edit, and share these definitions and extract structured information for further analysis. Indeed, several of these features were missing or not reported in previous studies (Table 4).

The ability to adapt the documentation procedures to the needs of clinical teams is another important aspect that our platform implements. Creating custom templates achieves the following: i. potentiates the flexibility of data acquisition; ii. supports asynchronous data acquisition by multidisciplinary teams (multiple team members can create numerous coordinated templates); and iii. allows the creation of automated reports. The separation between clinical concerns and technical design allows the implementation of new requirements without reprogramming the entire solution. For example, suppose new requirements are in order. In that case, the teams can easily update the templates (or new complementary templates created) without changing the data model or loss in interoperability.

The physical environment where it was projected and developed is also meaningful. Having the opportunity to work on the solution inside the institutional university clinic network offered the chance to receive constant direct feedback to the developing team and the healthcare end users. The shared environment promoted an iterative framework development aligned with users’ requirements. Another essential factor of this setup relates to the direct user support approach. In particular, the team held several workshops and meetings during the clinical implementation phase, which was critical during the initial development of the templates. The IT department’s continuous support was crucial for coordinating the presented framework and the systems already in place. Together, these measures significantly contributed to mitigating the transition friction between platforms.

Several challenges were identified during the development of the framework, particularly regarding the strong emphasis on longitudinal assessment. The paradigm shift from free text to a structured solution composed of coded elements required adopting and familiarizing with terminologies such as SNOMED-CT. The study, interpretation, and integration of new tools and the adherence to strict terminologies represents a paradigm change. In this sense, creating a dedicated, multidisciplinary team focused on a single case study allowed the development of the prototype, now available in a real setting and ready to be used and expanded to other clinical and research areas.

A current downside of the proposed solution is the lack of integration between the output structured documents and the SNS EHR. Administrative data and the documentation created in our platform are not automatically associated. In this sense, data duplication and inconsistencies between systems are potential risks. One possible solution is to generate reports based on the proposed platform and attach these documents (e.g., pdf files) to the hospital EHR system.

There are several similar approaches to generic structured data entry [5,17]. However, the solution presented answers the specific requirements to design a complementary solution to the existing SNS EHR system.

### 4.2. Comparative Analysis of the Framework

Security and data privacy are crucial when dealing with healthcare data and are a priority in our study. Similar to other studies, we adopted the HL7 CDA standard as the structure for healthcare documents and SNOMED-CT as the terminology.

The ability to generate healthcare templates capable of recording healthcare data without changing database entities and architecture was one of the main focuses of this study. Moreover, adding rules retrievable during the filling process was another important feature implemented in our architecture. This feature significantly improves data quality without needing ad hoc reprogramming to adjust specific aspects of the templates, etc. Data science can provide new insights for healthcare, but support for data extraction is often required. Thus, we also aimed to provide a tool to extract data from the healthcare documents created within the system. The majority of the studies using EHR were implemented in a hospital setting. Table 4 presents a summary of previous work in comparison to ours.

### 4.3. Future Work

Large datasets of structured clinical data will soon become available based on the continuous improvement and increased adoption of structured documentation systems. Moreover, longitudinal and high diversity of imaging, genetics, biological, environmental, and lifestyle information from single individuals to large cohorts will be available to improve healthcare [34]. The use of this information for research purposes is expected to enhance the state-of-the-art regarding the characterization and knowledge of certain diseases and pathologic mechanisms. It is also likely that the data generated could empower the healthcare community with support decision tools aided by machine learning and artificial intelligent systems, ultimately improving healthcare services for everyone. Additionally, the decrease in data duplication, the ability to share data and collaborate in the data collection process, as well as the definition of a uniform set of terms for the same discipline or health issue are factors that likely improve data quality and, ultimately, the quality of healthcare services.

To this end, future work should focus on validating the data extraction module and interface with clinical research groups from other fields. This module must abide by current legislation in terms of data privacy and requirements of anonymization and be assessed by a data protection officer. Accessing large datasets of structured clinical data allows the research community to observe features only revealed when the dataset is large enough.

Usability testing in software applications in the healthcare industry is crucial for promoting clinical efficiency, reducing provider burden, and maintaining clinical teams’ engagement. The usability assessment involves the evaluation of effectiveness, efficiency, and satisfaction with which specific users, representative of the primary end user, can achieve a specific set of tasks in a particular environment. The users in our use case were directly involved in the development process, the definition of features, and the user interface layout. Future work involves the usability assessment on a broader audience.

To extend the data sources, additional connections will be developed to third-party servers (e.g., imaging databases from national and international infrastructures, as well as biobanks).

## 5. Conclusions

The proposed system represents a novel approach to obtain structured clinical documentation in real-time encounters (consultation) that may require fast access to complex information, as was recently appraised by the COVID-19 pandemic. The approach allows the creation and storage of structured information, based on re-usable templates in a daily life clinical setting. Integrated with the previously established EHR system, the solution facilitates data recording and codifying clinical information in real-time. Through reporting structures also available to the users, the solution allows the creation of summaries of the patients’ clinical history, assisting the clinicians in producing relevant clinical documents, such as discharge notes.

The proposed solution enables a change in the current practices of the host University Hospital and will guide the clinicians to create enhanced clinical documentation and improve communication and interoperability. Ultimately, the availability of such dataset’s paves the way for research community collaborations, such as transnational collaborations in which the team is involved, allowing new approaches that were previously not attainable and fostering translational research.

In conclusion, the current solution improved the current EHR system used in our clinical setting (CHUC) by allowing the digital collection of structured and standardized clinical data using the SNOMED CT concepts in medical encounters for neuropediatric healthcare doctors of UNDA. Additionally, it allowed the definition of documentation architecture based on the definition of templates according to the HL7 CDA standard, enhancing future queries to the information stored. This potentiates the longitudinal view of the data, which is critical in a pediatric setting, supporting cohort selection and collaborative multidisciplinary studies.

## Figures and Tables

**Figure 1 healthcare-11-00973-f001:**
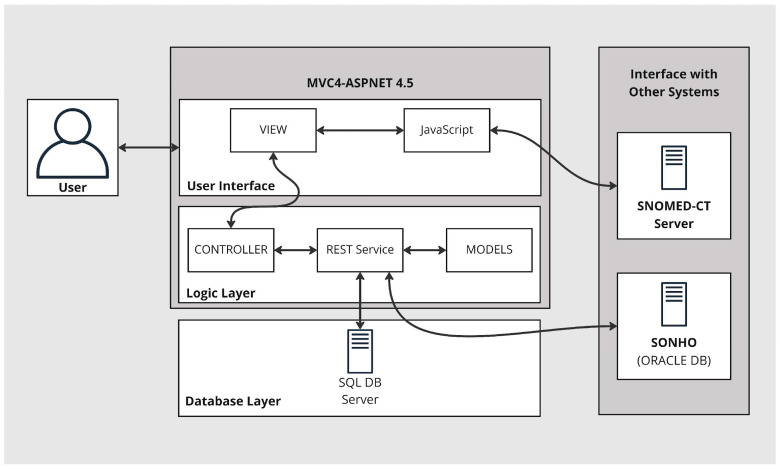
The architecture of the proposed solution. A three-layered system to extend the SClínico EHR and interfacing with a SNOMED-CT server and SONHO database.

**Figure 2 healthcare-11-00973-f002:**
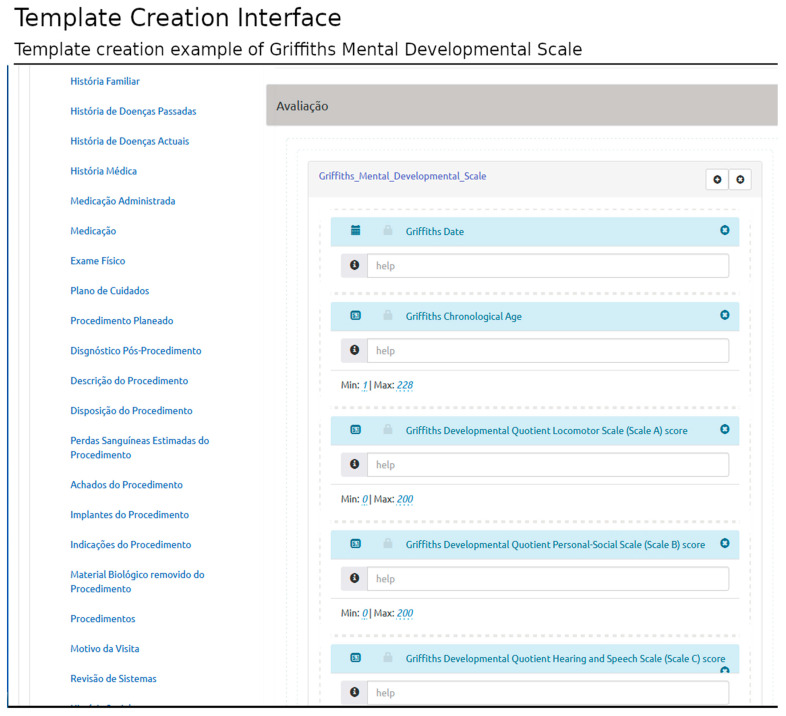
Example of template creation interface. In this case, the template creation of the Griffiths Mental Developmental Scale is shown. The template is included in the assessment section, named “Avaliação” in the Portuguese language. Other possible sections to capture healthcare information are listed at left (top to bottom English translation: Family History; History of Past Diseases; History of Current Diseases; Clinical History; Medication Administered; Medication; Physical Exam; Care Plan; Planned Procedure; Post-Procedure Diagnosis; Procedure Description; Disposition of the Procedure; Estimated Blood Loss From the Procedure; Procedure Findings; Procedure Implants; Procedure Indications; Biological Material Removed From the Procedure; Procedures; Reason of the Visit; and Systems Review).

**Figure 3 healthcare-11-00973-f003:**
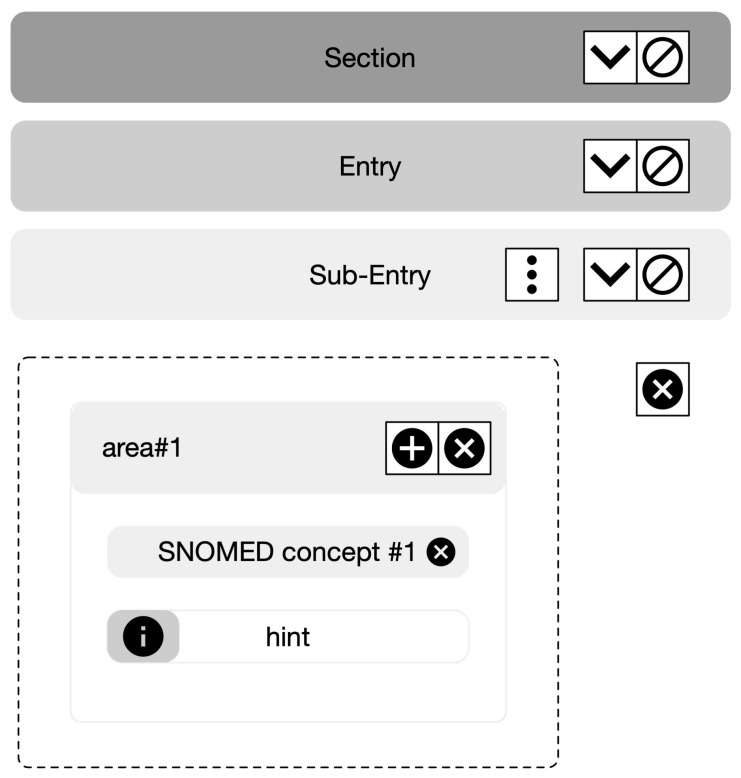
Mockup of the interface for template development. The HL7 CDA structure is graphically presented as follows: Section, Entry, and Sub-Entry are added to the template, as well as an area “area#1” with a single SNOMED-CT element “SNOMED concept”.

**Figure 4 healthcare-11-00973-f004:**
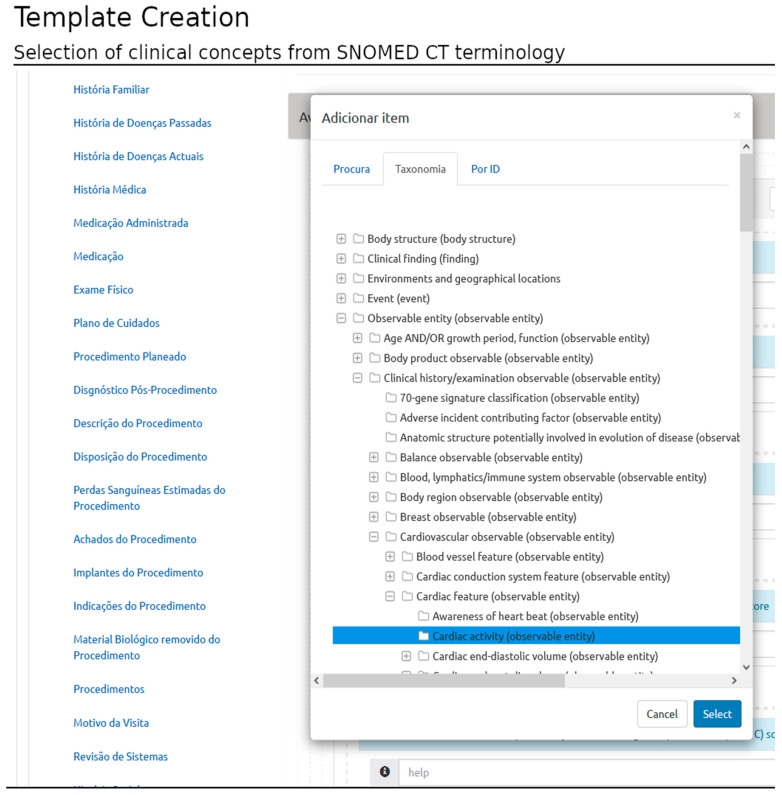
Template building: inclusion of clinical concepts based on the SNOMED CT standard. The user can select a SNOMED CT concept by either querying the terminology by a search word or a concept id or by querying the terminology tree. Possible sections to capture healthcare information are listed at left in the Portuguese language (top to bottom English translation: Family History; History of Past Diseases; History of Current Diseases; Clinical History; Medication Administered; Medication; Physical Exam; Care Plan; Planned Procedure; Post-Procedure Diagnosis; Procedure Description; Disposition of the Procedure; Estimated Blood Loss From the Procedure; Procedure Findings; Procedure Implants; Procedure Indications; Biological Material Removed From the Procedure; Procedures; Reason of the Visit; and Systems Review).

**Figure 5 healthcare-11-00973-f005:**
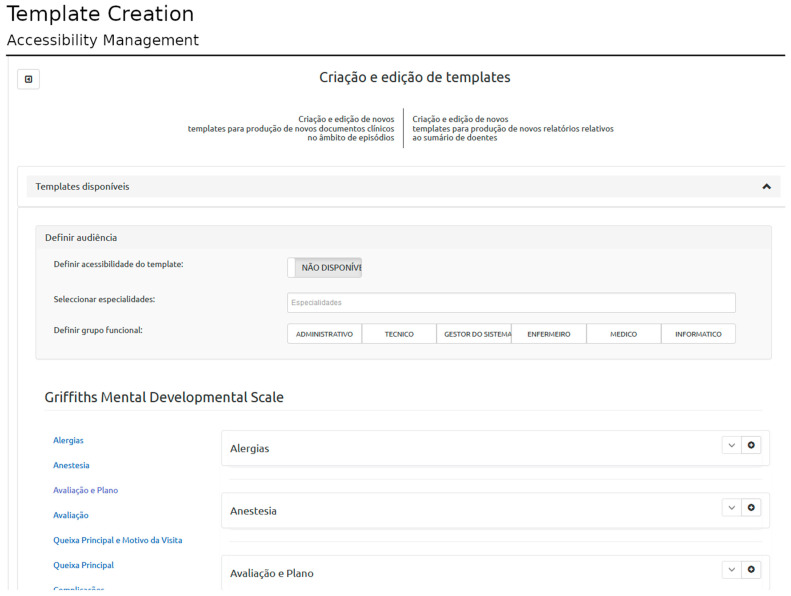
Template building: accessibility management. In this section of template building, the user is allowed to manage who will have access to the clinical research template. Here, it is possible to make the template available only for the author user or to share it within a specific medical specialty, making it accessible to all clinicians associated with that specialty. (The Portuguese terms have the following translations in English: Criação e Edição de Templates: Creation and Edition of Templates; Templates Disponíveis: Available Templates; Definir Audiência: Template Access Definition; Definir acessibilidade do template: Definition of template accessibility; Selecionar especialidade: Select speciality; Definir grupo funcional: Functional group definition; Administrativo: Administrative; Técnico: Technician; Gestor de Sistema: System Manager; Enfermeiro: Nurse; Médico: Doctor; Informático: Informatics; Alergias: Allergies; Anestesia: Anesthesia; Avaliação e Plano: Assessment and Plan; and Queixa Principal e Motivo da Visita: Main Issue and Reason for Visit).

**Figure 6 healthcare-11-00973-f006:**
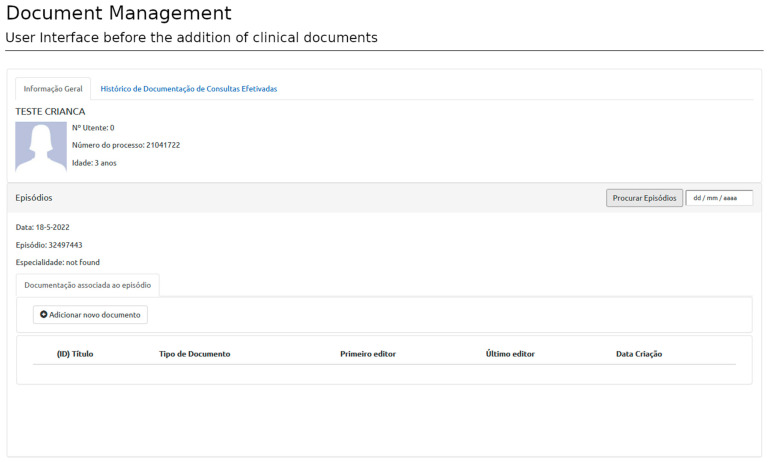
Document creation: User interface after authentication. The user is presented with the essential patient and episode information and can add clinical documents related to that medical encounter. This image describes a fake patient created for display purposes. (The Portuguese terms have the following translations in English: Informação Geral: General Information; Histórico de Documentação de Consultas Efetivadas: Documentation History of Medical Appointments; TESTE CRIANÇA: FAKE PATIENT; Nº Utente: User Number; Número do Processo: Process Number; Idade: Age; Episódios: Encounters; Data: Date; Episódio: Encounter; Especialidade: Speciality; Procurar Episódios: Look for Encounters; Documentação associada ao episódio: Documentation associated to the encounter; Adicionar novo documento: Add new document; (ID) Título: (ID) Title; Tipo de Documento: Document Type; Primeiro Editor: First Edited by; Último Editor: Last Edited by; and Data Criação: Date of Creation).

**Figure 7 healthcare-11-00973-f007:**
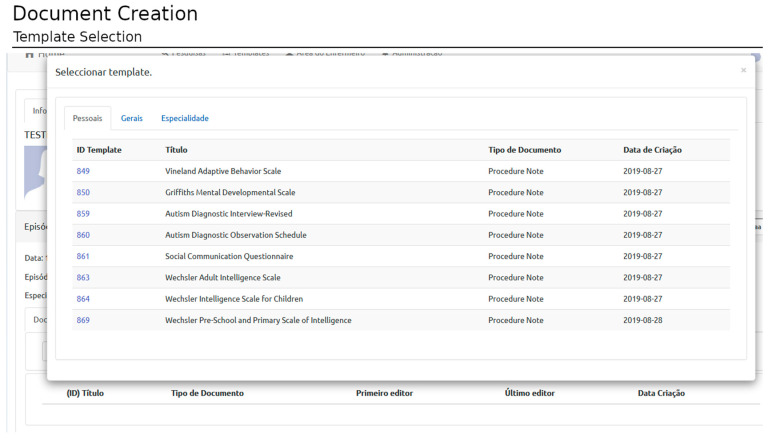
Document creation: Template selection. This window allows the user to select a template that is used to generate the clinical document structure, the clinical terminology, and the data types for each clinical term. This is relevant both for harmonized clinical practice and distributed research needs. (The Portuguese terms have the following translations in English: Selecionar Template: Select Template; Pessoais: Personal; Gerais: General; and Especialidade: Speciality).

**Figure 8 healthcare-11-00973-f008:**
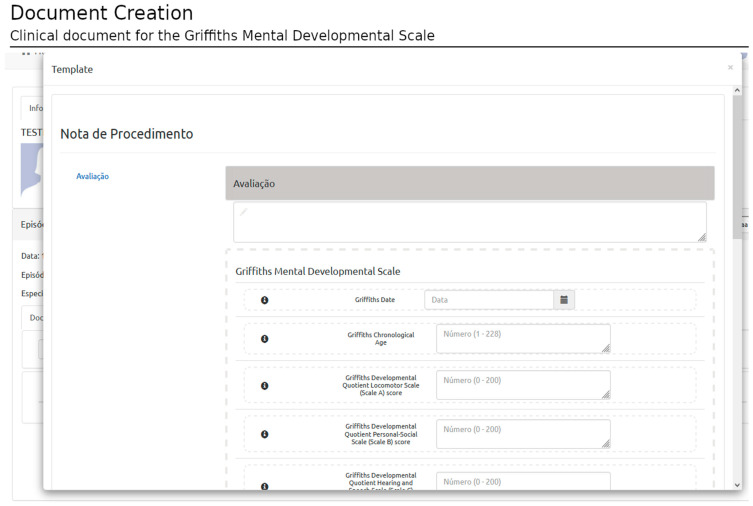
Document creation: Clinical document. This figure shows a clinical record for the Griffiths Mental Developmental Scale generated with the rules defined in the template of Figure 2. (The Portuguese terms have the following translations in English: Nota de Procedimento: Procedure Note and Avaliação: Assessment).

**Figure 9 healthcare-11-00973-f009:**
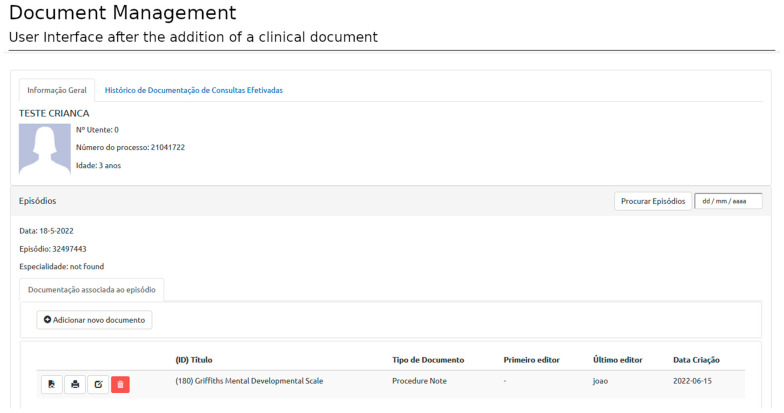
Document Management: User interface after the addition of a clinical document. After adding a clinical document to an episode, the user has the following management options: visualize; export a PDF; edit; or delete the document. Metadata about the document (document id, document name, type of document, document author, and date of creation) are also presented to the user. (The Portuguese terms have the following translations in English: Informação Geral: General Information; Histórico de Documentação de Consultas Efetivadas: Documentation History of Medical Appointments; TESTE CRIANÇA: FAKE PATIENT; Nº Utente: User Number; Número do Processo: Process Number; Idade: Age; Episódios: Encounters; Data: Date; Episódio: Encounter; Especialidade: Speciality; Procurar Episódios: Look for Encounters; Documentação associada ao episódio: Documentation associated to the encounter; Adicionar novo documento: Add new document; (ID) Título: (ID) Title; Tipo de Documento: Document Type; Primeiro Editor: First Edited by; Último Editor: Last Edited by; and Data Criação: Date of Creation).

**Figure 10 healthcare-11-00973-f010:**
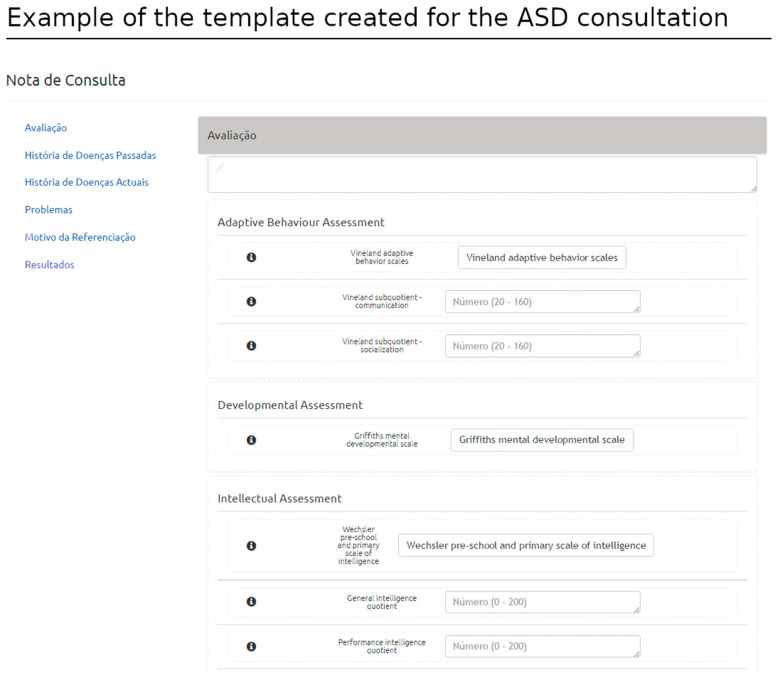
Example of the template created in the context of ASD consultation. Conceptual areas (adaptive behavior assessment, neurodevelopmental assessment, intellectual assessment) organize concepts while defining rules for data entry. (The Portuguese terms have the following translations in English: Nota Consulta: Consultation Note; Avaliação: Assessment; História de Doenças Passadas: History of Past Diseases; História de Doenças Atuais: History of Current Diseases; Problemas: Problems; Motivo da Referência: Reason for Referral; and Resultados: Results).

**Figure 11 healthcare-11-00973-f011:**
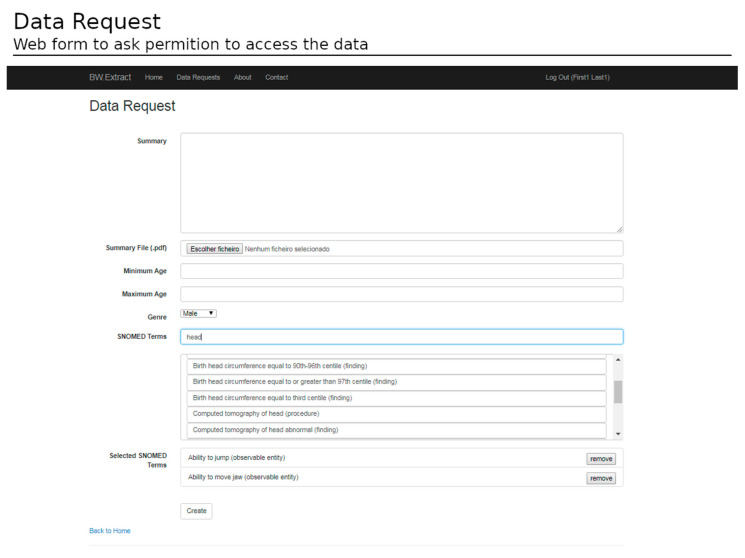
Web form for data requests. To ask for data, a researcher should provide a summary of the study and select data filters, such as the minimum and the maximum age, the genre, and the SNOMED CT concepts desired.

**Figure 12 healthcare-11-00973-f012:**
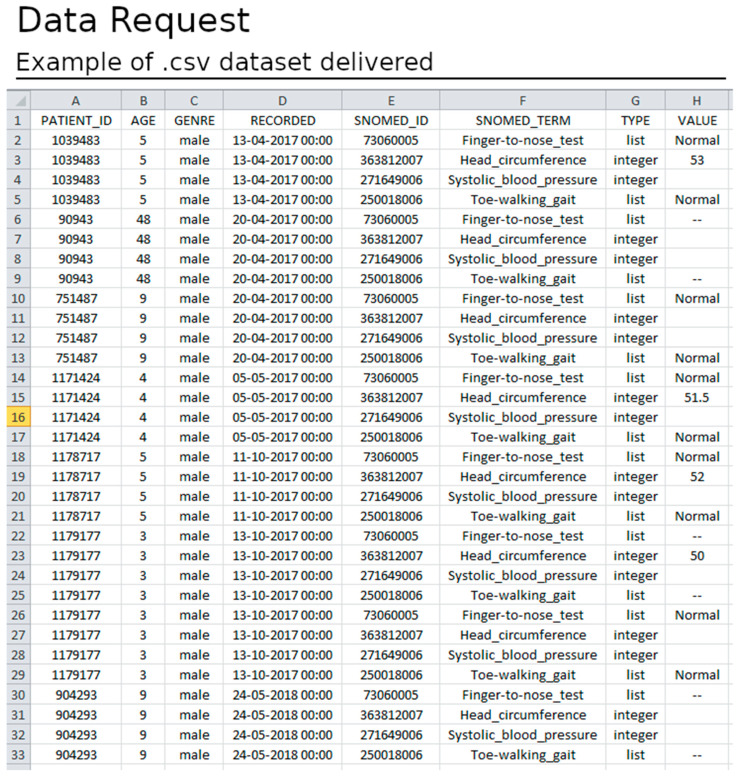
Example of a dataset obtained by a data request. The dataset is delivered in .csv long format (one record per line). The dataset is composed of several columns with metadata (the patient’s unique ID, the age of the patient, the sex of the patient, the date when the value was recorded, the SNOMED ID, the SNOMED definition, and the data type of the value) about the value recorded in the column VALUE.

**Table 1 healthcare-11-00973-t001:** Adopted function defined in EHR-S FM [23,24].

Id	Function Name	Relevant Tasks for This Study
DC.1	Care Management	It helps caregivers with record management, management of patient history, manage assessments, and provide measurements and results.
DC.2	Clinical Decision Support	It helps manage health information to provide decision support and provides support for population health investigations.
DC.3	Operation Management and Communication	Already provided by SClínico.
IN.1	Security	Already provided by SClínico.
IN.2	Health Record Information and Management	It helps with storing, managing, and extracting structured health record information.
IN.3	Registry and Directory Services	Already implemented by SClínico.
IN.4	Standard Terminologies and Terminology Models	It supports semantic interoperability by using standard terminologies, standard terminology models, and standard terminology services.
IN.5	Standard-based Interoperability	It supports operating seamlessly with other internal or external systems that adhere to recognized interchange standards and manage structured documents.
IN.6	Business Rules Management	Already provided by SClínico.
IN.7	Workflow Management	Already provided by SClínico.
S.1	Clinical Support	It supports the collection and distribution of local healthcare resource information through interactions with other systems, applications, and modules. Furthermore, it uses provider information to verify that a practitioner can use or access authorized data.
S.2	Measurement, Analysis, Research, and Reports	It supports the capture and export or retrieval of data necessary for report generation and ad hoc analysis.
S.3	Administrative and Financial	Already provided by SClínico.

**Table 2 healthcare-11-00973-t002:** The achieved functionalities [23,24].

Id	Function Name	Achieved?
DC.1	Care Management	Yes, caregivers can view, manage, and record patient data and access patient history, measurements, and results.
DC.2	Clinical Decision Support	Yes, healthcare givers can access health information useful for decision support, and record data for future population health investigations.
IN.2	Health Record Information and Management	Yes, users can store, manage and extract structured health record information.
IN.4	Standard Terminologies and Terminology Models	Yes, it implements the HL7 CDA standard to structure healthcare documents and SNOMED CT terminology to define healthcare terms.
IN.5	Standards-based Interoperability	Yes, it is achieved by implementing the HL7 CDA standard and SNOMED CT terminology.
S.1	Clinical Support	Yes, it is integrated within the SClínico platform and interacts with the SONHO database. Furthermore, it uses authentication to verify that a practitioner can use or access authorized data.
S.2	Measurement, Analysis, Research and Reports	Yes, users can generate reports in pdf or extract data for ad hoc analysis.

**Table 3 healthcare-11-00973-t003:** The addressed challenges in the implementation of the current solution [33].

Theme	Category	Sub-Category	How Was It Addressed in This Study?
Effects	Work for Healthcare providers	Efficiency	Less time-consuming tasks related to acquiring and managing paper-based records. Decreases the time needed to retrieve information in EHRs and reduces documentation time, for example, by using templates.
		Communication	Instant healthcare information coding in EHRs increased access to patient information through EHRs, enhancing communication within the healthcare team.
		Work Organization/Workflow	The use of predefined templates contributed to caregivers relying less on memory or written notes to keep track of which tasks had to be done.
		Workload	Decreased workloads as the adoption of the EHR improved communication, as well as the availability and accessibility of medical records.
		Support disease and quality management	The systematic storage of information in EHR supports better disease management.
		Support learning and decision-making	Not in scope.
	Data and information	Accessibility	Improved access to patient information and records as time needed to access information decreased. Furthermore, increased accessibility by allowing instant access to patient records.
		Data quality and accuracy	Enabling the capture of detailed data and improving documentation quality. Moreover, process-related and structural elements in EHR documentation were more easily retrieved than paper-based records. Patient data in EHR were highly valued as they contribute to more accurate data.
		Data storage and backup	Allowed convenient and systematic storage of data and information as digital records in EHRs were stored on servers with backup, reducing the likelihood of the data being lost.
	Care for patients	Quality of care	The use of templates and standardized medical terms prevents errors and improves patient safety. EHR allows caregivers to respond quickly to care needs, provide person-centered care, and carry out better follow-up care.
		Communication	EHR improves the clarification of information for patients.
		Patient empowerment	Increased patients’ access to full or partial medical records with the adoption of EHR.
		Change in time spent for patients	The use of templates and standardized medical terms helps reduce the time patients spend in the consultation because the information from previous consultations (even in other medical specialties) can be seen and used.
	Economic impact	Productivity	Not in scope.
		Decreased cost	Not in scope.
		Increased revenue and reimbursement	Not in scope.
Barriers	Support for users	Training and technical support	Several workshops and meetings were organized by the team in order to provide the best training and technical support. Furthermore, helpdesk services were created and are still available.
		User involvement	End user involvement during the planning, development, and implementation phases of the system life cycle of the EHR.
		Literacy and skill in technology	Provided according to the user’s specific needs during workshops, meetings, and through helpdesk services.
	EHR system	System integration and interoperability	HL7 CDA for structural interoperability and SNOMED-CT for semantic interoperability following the EHR-S FM framework.
		Trust and belief in EHRs	The system interoperability and integration with systems already used by caregivers, as well as increased data privacy and reduced risk of data loss contributed to increased trust in the current solution.
		System quality	The system compatibility, efficiency (fast responses), simple functionality, and straightforward user interface contributed to the quality of the system. Moreover, the use of HL7 CDA and SNOMED CT standards, as well as the ability to create templates and documents according to them, enhanced the flexibility of the system to be extended to other medical fields.
	Data and information	Privacy and security of data	User authentication using caregiver credentials ensured authorized and appropriate access to patient information. This also enabled better privacy, security, and confidentiality of the patient’s data.
		Data quality and accuracy	Using HL7 CDA to structure documents and SNOMED CT capture healthcare terms.
		Other concerns	No other special concerns related to data and information to be declared.
	Others	Resource constraints	Funding available for system upgrades/maintenance. Access to all health caregivers to computers and the solution developed.
		Legal liability	Complaint with the latest regulations.
		Awareness	Actively promoted and supported by workshops and meetings with several health caregivers.
		Policy support	Direct administrative involvement in identification and coordination of resources to support the solution.

**Table 4 healthcare-11-00973-t004:** Summary of the previous work in comparison to our work.

Authors	Year	Standards	Allow Template Creation and Generation of New Documents According to Them	Authentication and Security	Data Privacy	Supports Data Extraction for Group Analysis and Research Studies	EHR Type
Direito et al. (our work)	2023	HL7 CDA, SNOMED CT	Yes	Yes	Yes	Yes	Hospital-Hosted System
Shanbehzadeh et al. [18]	2021	HL7 CDA, SNOMED CT	No	No	No	No	Hospital-Hosted System
Oliveira et al. [19]	2020	HL7 CDA, OpenEHR, SNOMED CT	Yes	Not stated	Not stated	No	Hospital-Hosted System
Declerck et al. [20]	2015	HL7 CDA, ICD 9, ICD 10, LOINC, SNOMED CT, MedDRA	No	Not stated	Yes	No	Simulated System
López-Nores et al. [21]	2012	HL7, OpenEHR, EN13606	No	Yes	Yes	No	Hospital-Hosted System

## Data Availability

Data sharing not applicable. Nevertheless, IRB approvals are available for this grant.

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
