# Peer review of "Design and Implementation of a Collaborative Clinical Practice and Research Documentation System Using SNOMED-CT and HL7-CDA in the Context of a Pediatric Neurodevelopmental Unit"

_healthcare, 2023, doi:10.3390/healthcare11070973_

Round 1

Reviewer 1 Report

The authors report on implementing an EHR system based on HL7 CDA and SNOMED CT. The manuscript presents the architecture and sample screenshots from the developed prototype that can be used for further studies and experiments. A concrete research question is not present, but the authors argue for the advantages of their concept. However, there is no proof or factual comparison with other implementations from the literature or existing implementations in the hospital. A user test to support their claims is not present. It is unclear whether the system has been tried in practice, and with what results. Additional experiments are needed to make evident the claims made in the manuscript.

Further, the target for the implementation is unclear. The authors write about pediatric units, ASD, and others. It is unclear how specific the concept and implementation are for these target groups (or whether the concept is generic, but applied to a specific use case in this manuscript). The authors also mention the Covid19 pandemic in an unclear context. My suggestion is to make it clear that this is a case study of a generic concept applied to certain targets; please consider changing the title of your manuscript to clarify that this is a case study.

Please justify the use of SNOMED CT, as it is somewhat disputed in medical practice and not introduced in all countries.

Line 144: unclear what you mean by "real-time encoding" in this context.

The manuscript contains a large number of acronyms and names. Please add a table of acronyms at the end of your manuscript, to ease readability.

Please use a line drawing in Figure 1, as the shaded colours make the figure difficult to read. Please also use larger fonts.

Figures 2, 4, and onward are very difficult to read, due to unclear small fonts. Further, as the terms in the screenshots seem to be in Portuguese, please give some kind of translation for the relevant terms. As it is now, most of the figures only report the existence of some kind of user interface that is implemented using certain user interface primitives. Please also indicate what the differences are to the current implementations of such functionality.

Line 258: Please add "the" in front of SNOMED CT.

Unclear what Figure 5 shall show to the reader (as the concept of selecting among five different data types is rather basic).

Line 288:  Please define clearer what an "episode" is.

Lines 291ff: unclear

Figure 7: Authentication is important, but such functionality is beyond the main subject of your manuscript. I am unsure whether this deserves a separate figure. Figure 10 seems to be quite similar, but some steps later in the workflow.

Line 347: Unclear. What do you mean by "used as a routine"?

Line 356: Some product names are used that are not introduced.

The manuscript does not report on experiences of using the prototype in practice. Is there any functionality that users want in addition? How do the users cope with the user interface? Do they have comments on the use? I cannot find data from user testing or other analysis (such as the SUS). There is no comparison with other systems, prototypes, or previous implementations/routines. How much faster or otherwise measurably better is your prototype than other implementations? How does the medical staff react to the new interface?

The discussion is based on a generic discussion. It needs to be more concrete. For example, in Line 417, it is unclear what the requirements expected by the users are. Also Line 428ff: Please be more precise and describe the challenges outlined here.

The conclusion is rather generic. I don't understand Line 465, and how this relates to your manuscript. Further, Line 478ff: Please rewrite without using bullets. 

Why is the text in Appendix A in the appendix? Please integrate it into the main text. Please note: Figures 12 and 13 are not readable, thus, incomprehensible to the reader.

Reviewer 2 Report

Dear Authors,

After reviewing all the sections of this article, I have decided to recommend a major revision since there are many technical shortages and writing errors in the current version that must be revised significantly before making the final decision. Please consider the following comments while revising your manuscript.

1) The current abstract only focuses on what services your framework offers. Some details concerning the services (e.g., it allows queries on population statistics that are key for supporting the definition of local and global policies, which importance is now emphasized by the COVID pandemic) must be removed, and instead, you should include some sentences regarding what are the innovations of your framework and results compared to other existing frameworks.

2) There are several formatting and writing errors. I suggest asking an expert to proofread the paper once again carefully. For instance, on page 2, line 74, there are several extra spaces between sentences and two (..) at the end of a sentence. A sample of a wrong sentence is (... The ability to process and extract data from EHR in order to provide knowledge to the clinician is..) are you sure (clinician) word is correct here? Doctors are allowed to have access to EHR details, not all clinicians.

 3) This paper lacks a comparative analysis of the existing PHR/EHR management systems. I suggest citing related works and including a table and some plots to support your contributions. The current version does not provide enough analysis to prove your claims compared to other existing frameworks.

Reviewer 3 Report

This paper presents a collaborative platform between clinical and research teams for documenting clinical research using the HL7 CDA structured information model, clinical terminology (SNOMED CT) and the electronic health record (EHR). This approach enables real-time encoding and retrieval of clinical information while simultaneously querying the user for population statistics that are key to supporting local and global policy-making.

It's a worthy job. However, in my estimation, some weaknesses were identified, mainly in the presentation of the project. 

The following observations are detailed below:

A “Related Work” section should be added to present other relevant work.

What are the innovative elements of this work in relation to relevant pre-existing works e.g. openEHR.

Some typos to be corrected e.g., “….about the patient [1].”, “… measures [3].” “…healthcare challenges.” etc.

There is no related reference in the text: “The solution presented here is available within the SClínico platform” (ln 203) in the manuscript.

Also, there is no related  references for HL7 CDA, SNOMED Clinical Terms , etc.

Moreover, there is no citation in the manuscript for the Appendix A.

The figures showing the proposed platform are displayed in the Portuguese language.There is no English version? And if not, is there any intention to create one in English?

thank you

Good luck

Reviewer 4 Report

strength:
1.In Innovative aspects: The work effectively meets the requirements of semantic interoperability(SI) . HL7 CDA can meet the requirements of "The information model" , and SNOMED CT meets the needs of terminology.
2.There are detailed pictures in the paper, which effectively introduce the structure and interface of the system.
weakness:
1.Lack of introduction to relevant work: this article uses the structure of HL7 CDA and SNOMED CT, but does not introduce these work. This shows that the logic of the paper is not rigorous enough.
2.Lack of demonstration of the rationality of the method: In section 2.3, it is simply mentioned "the user can search for the best match". However, it does not introduce what is the best match and how to deal with unmatched cases
3.Discussion about result also needs to be improved: The paper only describes the various advantages of the system and lacks quantitative indicators. The five points listed in the penultimate paragraph of Introduction can be used as indicators for relevant personnel to score and compare with the original system
4.In writing: The excessive use of long sentences has brought some obstacles to reading. At the same time, in the penultimate paragraph of Introduction, there should be one ", " after "To implement and validate the new solution".

Reviewer 5 Report

Title: "Design and implementation of a collaborative clinical practice  and research documentation system in the context of a Pediatric Neurodevelopmental Unit"

In this article the authors proposed an approach for obtaining structured clinical documentation in real-time consultation that require fast access to complex information.

Even though the theme of the article is interesting and a lot of research currently going on to cope with current and future challenges. However, some moderate changes should be incorporated for acceptance and possible publication.

1.      The Title of the article should be informative i.e., it is required to show the methods used for some purpose/aim.

2.      The abstract should clearly present the limitations of classical documentation strategies and which one of them have been addressed in the proposed framework.

3.      It is also required to show in abstract that which parameters are used for the analysis and validation of the proposed work.

4.      Numerous punctuation mistakes are found in abstract i.e., Line No. 38, 53,61, 75, 77, 79, 87, 91,  & 93. These must be rectified.  

5.      In line 98:  the numeric value 66.500 does not have any unit to understand its meaning. Why?

6.      From 103 to 108 a complete paragraph is written in a single sentence. Which violates the rules of English reading semantics.

7.      In general, the introduction section should be re-arranged to maintain the chronological order of research cited and provide coherency and consistency in reading.

8.      A theoretical comparative analysis table should be added before the materials and methods section to show the existing frameworks in terms of  their strengths, and their weaknesses.

9.      In materials and methods section Figure 1. Should be clearly divided in two sections i.e., A & B….instead of Figure 1 & Figure 1B… As referenced in Line: 141-142.

10.   The architecture of proposed work presented in Figure 1 should be according to the standard reference model of EHR systems... which consist of  “Database, Workflow tools, Medication Management, Interface to other Systems (for interoperability), and User interface”. Some aspects are missing in the proposed model.

11.   In the results section “Due to language barrier I didn't understand the meaning of navigation labels given in figure 2,4,5”. These should be checked by their native language persons.

12.   The authors claim that they enhanced the clinical documentation, improved communication & interoperability and achieved high security and privacy of data. However, neither the results nor the discussion section presents that how the proposed system outperforms. This should be justified with solid experimental observations and analysis.  

13.   More, I believe that it will make this paper stronger if the authors add the smart features like alerts, notifications & summarized plots/ graphs to the target users and present their insightful implications.

14.   Moreover, it is suggested to search for latest published articles for citations and references as the cited papers are not sufficient to justify the claim of problem addressed.  

Round 2

Reviewer 1 Report

The authors have addressed my concerns sufficiently.

Some recommendations:

Line 685: please remove "Nevertheless,"

I did not find user testing being performed on the interface. The manuscript describes the evaluation of the functionality, but not the usability. Some form of usability evaluation should be performed (as also outlined in my previous review). I recommend performing user testing in a new manuscript and mentioning this as future work to be performed (in Section 4.3). As a note, the usability of healthcare systems is an important issue, as many healthcare workers are unable to understand advanced functionality of the healthcare systems.

The content of Figures 2, 4-11 is not of good graphical quality (too tiny letters, unsharp letters, etc.). It is hard to grasp some of the content. Further, there is also much unused white space. I recommend that you take some graphical measures to create screenshots that are readable.

Author Response

Reviewer 1

The authors have addressed my concerns sufficiently.

We thank the reviewer for the positive feedback.

Some recommendations:

Line 685: please remove "Nevertheless,"

Response: We have revised the manuscript accordingly.

I did not find user testing being performed on the interface. The manuscript describes the evaluation of the functionality, but not the usability. Some form of usability evaluation should be performed (as also outlined in my previous review). I recommend performing user testing in a new manuscript and mentioning this as future work to be performed (in Section 4.3). As a note, the usability of healthcare systems is an important issue, as many healthcare workers are unable to understand advanced functionality of the healthcare systems.

Response: We agree with the reviewer on the value of user usability in healthcare systems, its assessment, and evaluation. We also agree that user testing may be the focus of future work. The manuscript presents the platform and a specific use case - the platform was developed in coordination with one particular clinical team, fully engaged in the development. In this sense, a broader audience (not involved in the development process) for user testing will provide more comprehensive feedback on the strengths and weaknesses of this platform.

We have revised the manuscript accordingly:

page 26. “Usability testing in software applications in the healthcare industry is crucial for promoting clinical efficiency, reducing provider burden, and maintaining clinical teams' engagement. The usability assessment involves the evaluation of the effectiveness, efficiency, and satisfaction with which specific users, representative of the primary end-user,  can achieve a specific set of tasks in a particular environment. The users in our use case were directly involved in the development process, the definition of features, and the user interface layout. Future work involves the usability assessment on a broader audience.”

The content of Figures 2, 4-11 is not of good graphical quality (too tiny letters, unsharp letters, etc.). It is hard to grasp some of the content. Further, there is also much unused white space. I recommend that you take some graphical measures to create screenshots that are readable.

Response: We replaced Figures 2, 4-11 with more readable quality. However, we recognized that some content may be difficult to visualize in the manuscript and to address that we attached a compressed folder with all figures used in the manuscript with 400 dpi. In this folder, all figures can be opened in an image viewer and its content is readable when zoomed in without quality loss. We hope that this folder could be included in the final version to mitigate this problem.

Reviewer 2 Report

Dear Authors,

After reviewing all the sections of your article again, I have recommended a major revision since you did not consider almost half of my former comments. Please consider the following remarks carefully while revising your article.

1) Many formatting and writing errors are left in the current R1 version. For instance, on page 19 (section 3.5), line 547, "Functionalities tested focused on ....) how do you use two verbs beside each other? What is this language structure here? All the sections must be proofread by an expert or language editing service.

2) The tables added to the comparative analysis are too messy and unclear. You should define some evaluation metrics and provide some provable evidence that your contributions provide superior performance. What I see does not prove sufficient contributions in the area.

Author Response

Reviewer 2

Dear Authors,

After reviewing all the sections of your article again, I have recommended a major revision since you did not consider almost half of my former comments. Please consider the following remarks carefully while revising your article.

1) Many formatting and writing errors are left in the current R1 version.

Response: We have performed an additional proofreading assessment to remove any errors.

For instance, on page 19 (section 3.5), line 547, "Functionalities tested focused on ....) how do you use two verbs beside each other? What is this language structure here? All the sections must be proofread by an expert or language editing service.

Response: We have revised the manuscript according to reviewer’s comments:

page 20. “We performed functionality tests to ensure integration with other hospital EHR systems and objectively assessed the implementation of functionalities. Our tests focused on the functional components in EHR-S FM.”

2) The tables added to the comparative analysis are too messy and unclear. You should define some evaluation metrics and provide some provable evidence that your contributions provide superior performance. What I see does not prove sufficient contributions in the area.

Response: As highlighted by reviewer 1, the manuscript was more focused on the evaluation of the functionality. We agree with this assessment and further discuss the functionalities implemented based on the HL7 EHR-S FM standard (Table 2) and the challenges overcome with this solution based on the framework used in a scoping review dedicated to the assessment of implementation and barriers found when adopting EHR systems (Table 3).

Here, we aimed to critically discuss the functionalities implemented considering a standard description and common understanding of functions for healthcare settings. This community effort is informed by industry advances/directions, regulatory changes, learning from work from functional profiles, and participation by the international community.

Table 4 presents the comparison between our assessment regarding HL7 EHR-S FM implemented functionalities  and state-of-the-art platforms. We address our contribution in the area as the number of functions implemented and the strengths of our approach to collaborative work.

As recommended by reviewer 1, we emphasize that future work should focus on usability assessment and performance analysis, using a broader audience/group of testers. The development team should not be directly involved in the evaluation process, as they were engaged in the definition of requisites and features, and therefore, biased to usability assessment tests regarding effectiveness, efficiency, and satisfaction. Future work should address this limitation.

We have revised the manuscript accordingly:

page 20. “We performed functionality tests to ensure integration with other hospital EHR systems and objectively assessed the implementation of functionalities. Our tests focused on the functional components in EHR-S FM [23], [24]. This model has been widely reviewed by healthcare providers, vendors, public health agencies, regulatory and accreditation bodies, professional societies, trade associations, researchers, and other stakeholders and expresses consistent EHR system functionality. Table 2 summarizes the EHR-S FM functionalities implemented in the platform. Considering the main functionalities in Table 2, Table 3 describes the challenges faced while implementing our solution in the EHR Hospital-Hosted system following the EHR-S FM framework as a standard.”

page 26. “Usability testing in software applications in the healthcare industry is crucial for promoting clinical efficiency, reducing provider burden, and maintaining clinical teams' engagement. The usability assessment involves the evaluation of the effectiveness, efficiency, and satisfaction with which specific users, representative of the primary end-user, can achieve a specific set of tasks in a particular environment. The users in our use case were directly involved in the development process, the definition of features, and the user interface layout. Future work involves the usability assessment on a broader audience.”

Reviewer 3 Report

Good work!

Author Response

Reviewer 3

Comments and Suggestions for Authors

Good work!

Response: Thank you for your contributions that helped us to improve this manuscript

Reviewer 5 Report

Title: “Design and implementation of a collaborative clinical practice and research documentation system using SNOMED-CT and HL7-CDA in the context of a Pediatric Neurodevelopmental Unit”

I reviewed the revised version of above-entitled manuscript and found out that the authors satisfactorily incorporated all the required changes suggested by the reviewers in first round of review. Therefore, I hereby accept the manuscript for publication.

Author Response

Reviewer 5

I reviewed the revised version of above-entitled manuscript and found out that the authors satisfactorily incorporated all the required changes suggested by the reviewers in first round of review. Therefore, I hereby accept the manuscript for publication.

Response: Thank you for your contributions that helped us to improve this manuscript.
